# No effect of additional education on long-term brain structure, a preregistered natural experiment in thousands of individuals

**Nicholas Judd[1,2,3]\*, Rogier Kievit[1]**

[1]Cognitive Neuroscience Department, Donders Institute for Brain, Cognition, and Behavior, Radboud University Medical Center, Nijmegen, Netherlands; [2]Department of Psychology, Stockholm University, Stockholm, Sweden; [3]Swedish Collegium for Advanced Study, Uppsala, Sweden

## eLife Assessment

A regression discontinuity analysis finds essentially no effect of 1 additional year of secondary education on brain structure in adulthood. This is a **valuable** finding that adds to the literature on the impact of education on brain health. While the finding is **convincing** on its own, as the analysis was pre-registered and very carefully conducted, the impact is limited as the manipulated variable only relates to a single additional year of education (remaining in education to 15 vs 16 years of age).

**Abstract** Education is related to a wide variety of beneficial health, behavioral, and societal outcomes. However, whether education causes long-term structural changes in the brain remains unclear. A pressing challenge is that individuals self-select into continued education, thereby introducing a wide variety of environmental and genetic confounders. Fortunately, natural experiments allow us to isolate the causal impact of increased education from individual (and societal) characteristics. Here, we exploit a policy change in the UK (the 1972 ROSLA act) that increased the amount of mandatory schooling from 15 to 16 years of age to study the impact of education on long-term structural brain outcomes in the UK Biobank. Using regression discontinuity (RD) – a causal inference method – we find no evidence of an effect from an additional year of education on any structural neuroimaging outcomes. This null result is robust across modalities, regions, and analysis strategies. An additional year of education is a substantial cognitive intervention, yet we find no evidence for sustained experience-dependent plasticity. Our results provide a challenge for prominent accounts of cognitive or 'brain reserve' theories which identify education as a major protective factor to lessen adverse aging effects. Our preregistered findings are one of the first implementations of RD on neural data – opening the door for causal inference in population-based neuroimaging.

## Introduction

Access to education is codified as a fundamental human right with immense societal and economic benefits (*Geneva Convention IV, 1949*; *UNCRC, 1989*). Individuals who experience more education tend to show a wide variety of beneficial health, cognitive, and neural outcomes (*Lövdén et al., 2020*). Correlational evidence across disparate contexts, countries, and demographics offers overwhelming support for these findings (*Strand et al., 2010*; *Ross and Wu, 1996*; *Kobayashi et al., 2017*; *Waldstein et al., 2017*; *Livingston et al., 2020*; *Montez et al., 2012*; *Turrini et al., 2023*).

**\*For correspondence:**
nicholas.judd@su.se

**Competing interest:** The authors declare that no competing interests exist.

Education not only increases learned skills and knowledge, but also reasoning and fluid intelligence abilities (*Ceci, 1991*; *Ritchie and Tucker-Drob, 2018*; *Means and Voss, 1996*; *Judd et al., 2022*). One plausible mechanism underlying these widespread benefits from education is the long-term reorganization of brain structure. Lifespan theories on heterogenous neurodevelopment place particular emphasis on education (*Cabeza et al., 2018*; *Buckner, 2004*; *Stern, 2002*; *Lindenberger and Lövdén, 2019*; *Newcombe, 2011*) and correlative work further supports the role of brain structure as a mechanism (*Waldstein et al., 2017*; *Walhovd et al., 2022*; *Noble et al., 2012*; *Chan et al., 2018*). For instance, more educated individuals show higher mean cortical thickness (CT) in later life – taken as evidence of 'increased brain reserve' lessening the neurodegenerative effects of aging (*Kim et al., 2015*; *Liu et al., 2012*).

Yet, strong causal evidence for the effect of education on brain structure is severely lacking (*Seyedsalehi et al., 2023*). Both ethical and practical constraints mean that the effect of education cannot be experimentally tested. This makes it unclear if any findings from education are causal, or if instead, they reflect a complex web of preexisting, interacting, sociodemographic and individual characteristics (*Lövdén et al., 2020*; *Hart et al., 2021*) – many of which have been associated with brain structure (e.g. intelligence [*Jung and Haier, 2007*], parental income [*Farah, 2017*], neighborhood pollution [*Bottenhorn et al., 2024*]). Among the plethora of environmental factors, there is also a substantial (~40% heritability) genetic component implicated in educational attainment, further confounding potential effects (*Branigan et al., 2013*; *Lee et al., 2018*). In addition, access to higher education involves substantial selection processes at the level of the individual and educational system, which further complicate the causal pathways involved.

One solution to this problem is using a *natural experiment* – a 'random-like' (exogenous) external event – allowing causal inference in observational phenomena (*Lee and Lemieux, 2010*; *Cattaneo and Titiunik, 2022*). A crucial feature of this design is an assignment mechanism outside a participant's control, usually of natural (e.g. geographical, weather events) or governmental (e.g. policy decisions, cutoff rules) origin. For instance, a law that increases the number of mandatory school years affects everyone equally, in turn, decoupling individual characteristics and other unmeasured confounders from the effects of additional education (*Angrist and Krueger, 1991*). One of the most widely used causal inference analysis techniques is regression discontinuity (RD), which is applicable when treatment is assigned via a cutoff of a particular score or running variable (*Lee and Lemieux, 2010*; *Cattaneo and Titiunik, 2022*). Recent advances in the field of econometrics have optimized inference and largely standardized RD analysis (*Cattaneo and Titiunik, 2022*; *Armstrong and Kolesár, 2018*; *Armstrong and Kolesár, 2020*; *Cattaneo et al., 2019*). Similar natural experimental designs have provided robust evidence that additional education *causes* an increase in intelligence (0.14–0.35 SD; *Ritchie and Tucker-Drob, 2018*). Despite the many strengths and potential causal insights natural experiments offer the field of cognitive neuroscience, they have not been used to address such questions.

On September 1 1972, the minimum mandatory age to leave school was raised from 15 to 16 years of age in England, Scotland, and Wales (called here 'ROSLA'; *ROSLA, 1972*). The consequence of this law change was substantial: it resulted in almost 100% of children aged 15 staying in school for an additional year, in turn, increasing formal qualifications, income, and cognition (*Davies et al., 2018*; *Gorman, 2023*; *Clark and Royer, 2010*; *Clark and Royer, 2013*; *Barcellos et al., 2023*; *Barcellos et al., 2018*). However, whether this substantial intervention also affected the long-term brain structure of those born right around the cut-off remains an open question. This is unfortunate, as RD is a powerful tool to study phenomena that cannot (ethically or practically) be randomized. Recent developments of increased large population-based neuroimaging cohorts provide the sample size needed to make use of these cross-disciplinary methods. One such cohort, the UK BioBank, has recruitment criteria matching the geographic and birth window of the natural experiment ROSLA (*Alfaro-Almagro et al., 2018*). This provides the ideal opportunity to test, for the first time, the causal effect of a year of education on long-term structural neuroimaging properties.

Using a preregistered design (https://osf.io/rv38z/) with over 30,000 participants, we evaluate if an additional year of education, as mandated by ROSLA, causes changes in six global neural properties (total surface area [SA], average CT, normalized total brain volume [TBV], mean weighted fractional anisotropy [wFA], white matter hyperintensities [WMh], and normalized cerebral spinal fluid volume) as compared to individuals born before the cut-off. It is also possible that an effect from education

could manifest only in specific regions, in the absence of broad, macro-neural effects. For this reason, we further tested 66 individual cortical regions for SA and CT, 27 white matter fiber tracks for fractional anisotropy, and subcortical gray matter volume in 18 regions. Taken together, the combination of cutting-edge quantitative methods, a large sample, a well-validated natural experiment, and high-quality imaging allows us to examine, for the first time, if an additional year of education causes long-term structural reorganization in the brain.

## Results

To test if an additional year of education causes long-lasting changes in neural properties, we used fuzzy local-linear RD (*Armstrong and Kolesár, 2018*; *Armstrong and Kolesár, 2020*). This continuity-based technique exploits the fact that ROSLA affected individuals based on a date of birth (DOB) cutoff (September 1, 1957). More specifically, adolescents born after this date had to spend one more year in school than those born only one day earlier. Compliance with ROSLA was very high (near 100%; *Figure 1—figure supplement 2*). However, given the cultural and historical trends leading to an increase in school attendance before ROSLA, most adolescents were continuing with education past 15 years of age before the policy change (*Figure 3—figure supplement 2*). Prior work has estimated that 25% of children would have left school a year earlier if not for ROSLA (*Clark and Royer, 2010*). Using the UK Biobank, we estimate this proportion to be around 10%, as the sample is healthier and of higher SES than the general population (*Figure 1—figure supplement 2*; *Supplementary file 2*; *Fry et al., 2017*; *van Alten et al., 2024*; *Lyall et al., 2022*).

Local-linear RD analysis tests the effect of this policy change by comparing the limits of two non-parametric functions, one fit using only participants right before the policy change to another function fit on participants right after (*Cattaneo and Titiunik, 2022*). If an additional year of education affects neural outcomes, we assume that brain structure will be *discontinuous* exactly at the cutoff. In contrast, if ROSLA does not affect long-term neural outcomes, the functions should be *continuous* around the cutoff.

Using fuzzy local-linear RD, we fit a series of regressions to various measures of brain structure to test for any discontinuity at the cutoff. To determine the optimal number of participants to include, we use mean square error optimized bandwidths to a maximal range of 10 years before and after the cutoff (N>30,000). Doing so, we observed no evidence of an effect from additional education on any of our preregistered global neuroimaging measures (*p's*>0.05; *Supplementary file 2*). In other words, the relationship between the year of birth and neural outcomes was *continuous* around ROSLA's cutoff, indicating no differences in global structural measures from an extra year of education. The optimized bandwidths used in this analysis included participants born 20–35 months around the cutoff (average N=5124). The absence of a causal effect of education was observed for all our global neuro-imaging metrics: total SA, average CT, normalized TBV, mean wFA, WMh, or normalized cerebral spinal fluid (CSF) volume (*Figure 1*; *Figure 1—figure supplement 3*). These results did not change when imputing missing (~4%) covariate data.

These findings strongly suggest that an additional year of education did not lead to changes detectable by MRI decades later. However, to ensure the validity of our (causal) inferences, a critical step in any RD approach is to test the validity of the design (*Cattaneo and Titiunik, 2022*). For instance, if participants can manipulate their treatment by sorting around the cutoff, this severely limits the strength of causal claims. In the context of ROSLA, this is highly unlikely since the assignment was based on DOB (*ROSLA, 1972*; *Geruso and Royer, 2018*). However, for completeness, we tested this question (*Cattaneo et al., 2018*), finding no evidence that individuals were somehow able to adjust their enrollment ($T_q$ = –0.72, p=0.47; *Figure 1—figure supplement 2*). A second validation approach is to employ *placebo outcome tests* (*Cattaneo and Titiunik, 2022*). This approach uses variables that should *not* be causally related to your treatment (e.g. an additional year of education), to ensure the absence of spurious effects arising through unknown mechanisms. To accomplish this, we used all of our neuroimaging covariates (e.g. sex, head motion, site) as placebo outcomes, under the assumption that ROSLA should not affect these variables. Other than one covariate (summer), which was deterministically related to ROSLA (therefore excluded), none of the placebo outcomes were related to ROSLA (*Supplementary file 1*). Our findings add to the existing body of prior work (*Davies et al., 2018*; *Clark and Royer, 2010*; *Clark and Royer, 2013*; *Barcellos et al., 2018*) further solidifying ROSLA as a valid natural experiment.

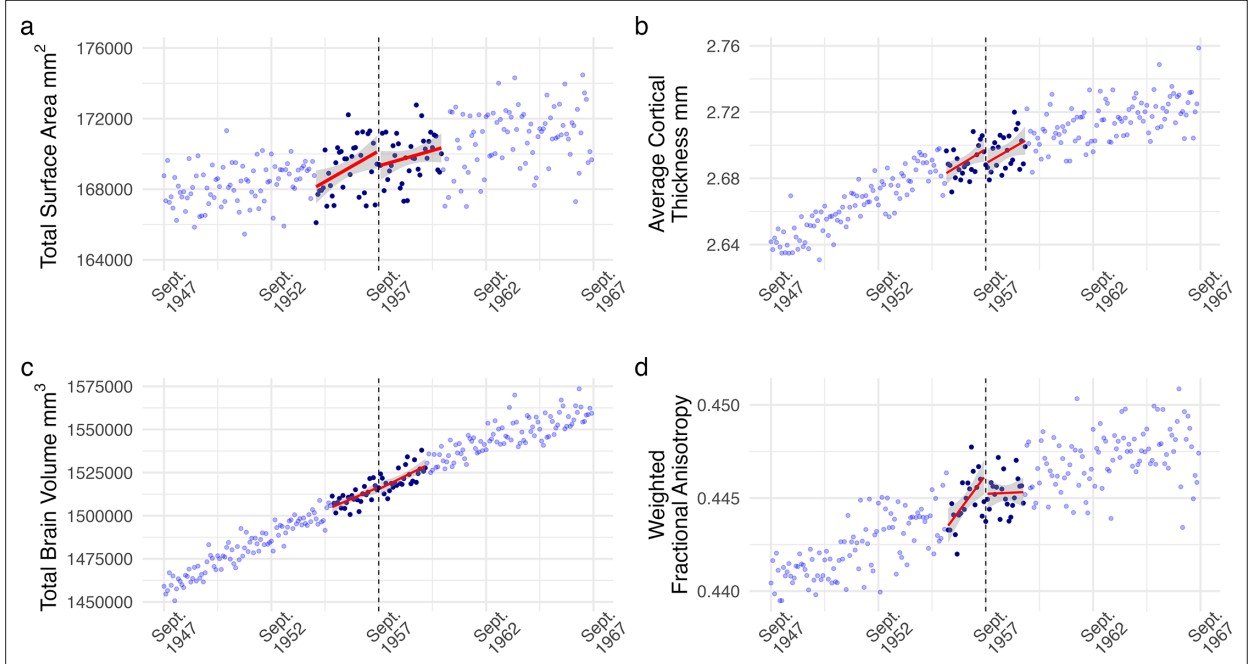

**Figure 1.** Regression discontinuity (RD) plot of monthly averaged (**a**) total surface area, (**b**) average cortical thickness, (**c**) total brain volume, and (**d**) weighted fractional anisotropy plotted by the participant's date of birth in months (our running variable). Each dot reflects the average value for individuals born in that month. The dashed line corresponds to September 1957, the date of birth inclusion cutoff for an additional year of mandatory education from ROSLA. Dark blue dots represent all individuals within the mean-squared error-optimized bandwidths, reflecting participants used for analysis with a local-linear fuzzy RD approach. We found no evidence of an effect from an additional year of education on any of the preregistered structural neuroimaging measures across various analysis specifications (***Supplementary file 2–4 and 6***).

The online version of this article includes the following figure supplement(s) for figure 1:

**Figure supplement 1.** Raincloud plots illustrating the distribution of participants (**a**) date of visit, (**b**) date of birth (DOB), and (**c**) of the participant's age at neuroimaging (mean = 61.89).

**Figure supplement 2.** This regression discontinuity plot shows the effect of the law on the percentage of students staying an additional year in school (Y) by date of birth monthly averaged bins.

**Figure supplement 3.** Regression discontinuity plots of monthly averaged.

**Figure supplement 4.** Regression discontinuity plots of monthly averaged (**a**) total surface area, (**b**) cerebrospinal fluid volume, (**c**) average cortical thickness, (**d**) white matter hyperintensities, (**e**) total brain volume, and (**f**) weighted fractional anisotropy, plotted by the participant's date of birth in months (X; our running variable).

Next, we tested whether there may be any regionally specific neuroimaging effects. It is possible that an additional year of education caused localized neural changes that are not picked up globally. As preregistered, we used the Desikan-Killiany cortical atlas (***Desikan et al., 2006***) to test the effect of an additional year of education on 33 bilateral regions for both CT and SA. These analyses included on average 5080 effective participants (N range = 3884–7771) for CT and 5392 participants (N range = 3739–8727) for SA. Despite these relatively high participant numbers, we did not find an additional year of education to cause changes in any regions for CT or SA ($p's_{FDR} >0.05$). This was also the case for weighted mean fractional anisotropy in all 27 tracks tested (***de Groot et al., 2013***; $p's_{FDR} >0.05$; mean n=4766). Lastly, we tested the volume of 18 subcortical regions, finding none to be related to an additional year of education ($p's_{FDR} > 0.05$, mean n=5174). To summarize, there was no evidence of an additional year of education affecting any regional neuroimaging measures with fuzzy local-linear RD (***Figure 2—figure supplement 1***).

## Bayesian local randomization robustness analysis

As an additional preregistered robustness test, we used an alternate framework, often referred to as 'local randomization', to analyze natural experiments. This approach works under the assumption that individuals close to the cutoff are exchangeable and similar except for the treatment (in our

case an additional year of school; *Cattaneo and Titiunik, 2022*; *Cattaneo et al., 2024*). To implement this analysis, we compared participants born *exactly* right before the cutoff (August 1957; n ≈ 130) to those born right after the cutoff (September 1957; n ≈ 100). An additional benefit is that we implemented this analysis in a Bayesian framework, allowing us to more readily interpret the strength of evidence either in favor of the null or alternative hypothesis. Our preregistered default point null Bayes factors were too wide, arguably providing relatively strong evidence in favor of the null. Taking a more conservative approach, we report these analyses with a narrower normal prior (mean = 0, sd = 1). Doing so, we replicated and extended our findings from fuzzy local-linear RD analysis, observing *strong* evidence in favor of the null hypothesis for total SA ($BF_{01}$=18.21), average CT ($BF_{01}$=15.09), TBV ($BF_{01}$=13.61), wFA ($BF_{01}$=11.63), WMh ($BF_{01}$=13.26), and cerebral spinal fluid volume ($BF_{01}$=14.50). In addition, we tested across a range of priors which did not meaningfully affect our inferences, as each showed evidence in favor of the null (*Figure 3—figure supplement 1*; *Supplementary files 4 and 6*).

The two quantitative RD approaches described here (local-linear and local randomization) have strengths and challenges similar to the widespread bias versus variance tradeoff. As the number of months on either side of the cutoff increases, bias is introduced as participants become less similar.

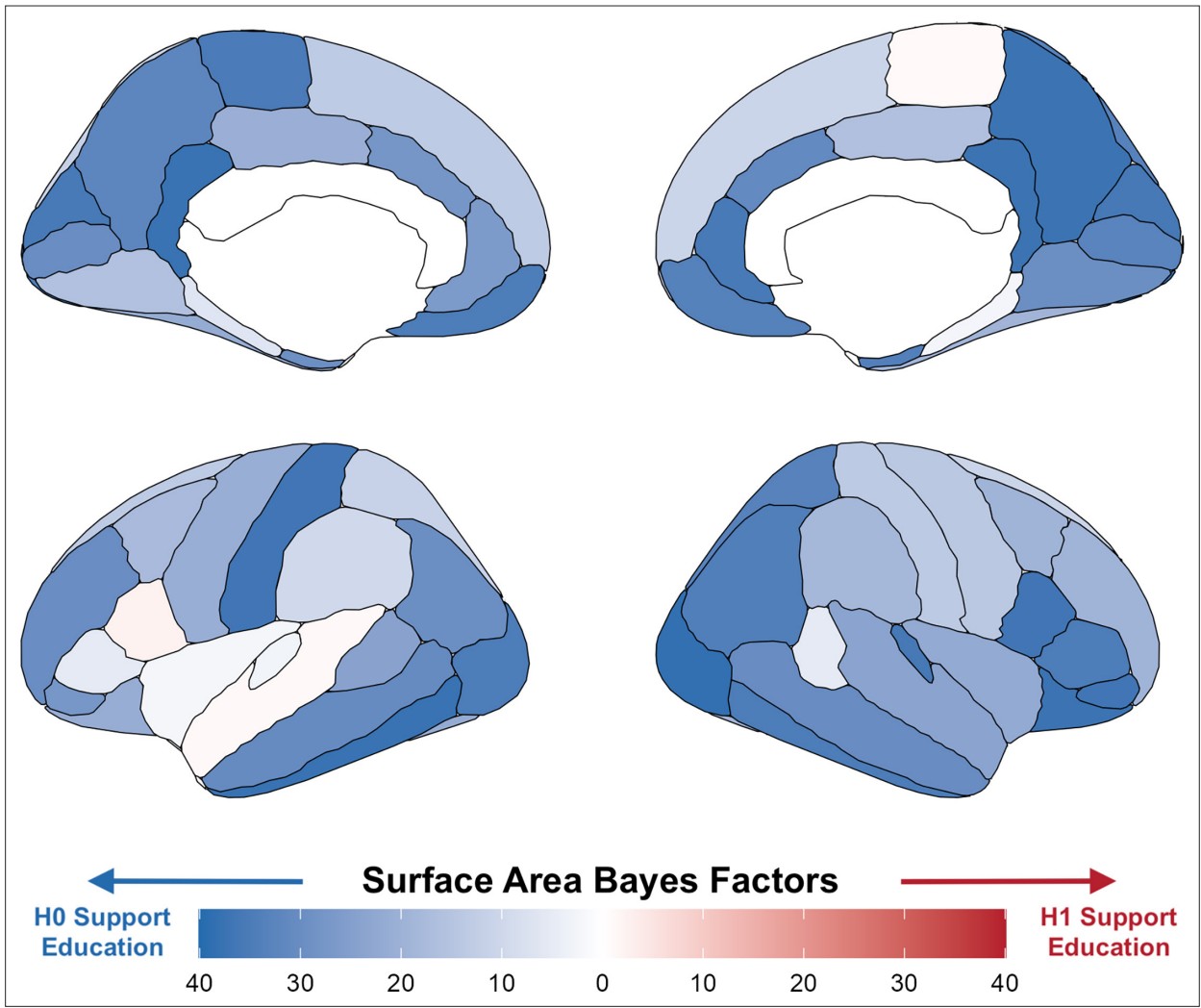

**Figure 2.** Bayes factors for surface area per region using a local randomization analysis with a 5-month window around the onset of ROSLA (September 1, 1957). Illustrating widespread evidence against the effect of a year of education on total surface area. The regionally specific analysis of these Bayes factors [reported prior: normal(0, 1)] was not preregistered and serves to illustrate our global neural findings.

The online version of this article includes the following figure supplement(s) for figure 2:

**Figure supplement 1.** Raincloud plots of (**A**) the effective number of observations and (**B**) uncorrected p-value of a local-linear fuzzy RD per region per modality [cortical thickness (CT), Surface Area (SA), Subcortical regions, and weighted mean fractional anisotropy (wFA)].

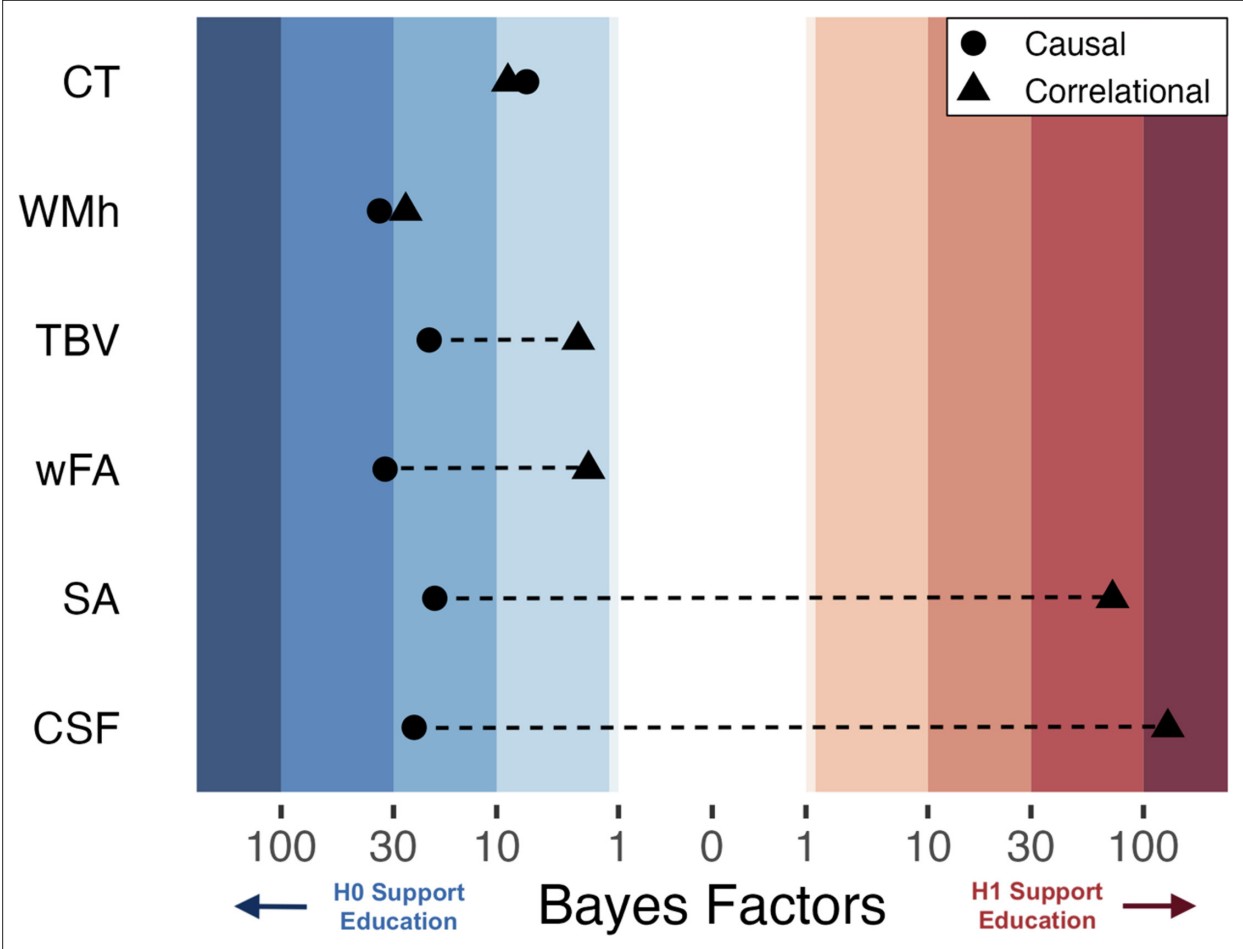

**Figure 3.** Bayesian Heat Plot: Bayesian evidentiary strength (x-axis) for causal (circles) and correlational (triangles) estimates reflecting the impact of one additional year of education on global neuroimaging measures (y-axis; average cortical thickness [CT], white matter hyperintensities [WMh], total brain volume normalized for head size [TBV], mean weighted fractional anisotropy [wFA], total surface area [SA], and cerebral spinal fluid volume normalized for head size [CSF]). Stripped bands reflect the strength of evidence using Jeffrey's criteria 1961. For the causal estimates, positive Bayes Factors indicated support for the alternative hypothesis that an additional year of education affects the brain, while negative values indicate support for the null hypothesis of no effect. The causal and correlational parameters come from the same set of participants (n ≈ 1200) born from April 1957 until Jan 1958. The causal parameter is an estimate of the effect of ROSLA with a 5-month window local-randomization analysis. The correlational parameter is an estimate of the association between a participant's self-reported educational attainment in years and (global) neuroimaging measures. Note: The association between educational attainment and CSF did not hold across eight replication sub-cohorts, yet surface area demonstrated a robust association (***Figure 3—figure supplements 2 and 3***).

The online version of this article includes the following figure supplement(s) for figure 3:

**Figure supplement 1.** An overview of the Bayesian priors used in our Bayesian local-randomization analysis.

**Figure supplement 2.** An overview of eight cohorts/subsets (plus the original 5-month local randomization cohort; s1957) spaced 2 years apart with 10-month windows of included subjects in panel **a**.

**Figure supplement 3.** Illustrates the posterior of the associational effect of educational attainment on neuroimaging measures across nine 10-month subcohorts of UK Biobank data spaced 2 years apart.

Yet at the same time, the sample size increases, thereby lowering the variance of the estimate. Our local randomization approach runs a negligible risk of bias, but at the cost of a relatively modest sample size: One could argue the 230 participants included in our 1-month window are too few for neuroimaging outcomes. To examine the consequences of this tradeoff, we therefore expanded the boundary to a 5-month window around September 1, 1957 (n per group ≈ 600). This larger participant pool provided further evidence in support of the null hypothesis of education not affecting global neural measures (***Figures 2 and 3***, ***Supplementary file 6***). Lastly, we repeated our placebo outcome tests for both a 1- and 5-month window local randomization analysis, finding no associations

(*Supplementary file 5 and 7*), demonstrating the robustness of the natural experiment ROSLA and our analysis approach.

## Correlational effect of education

The above analyses employed an RD design allowing us to investigate the hypothesized causal effect of (additional) education on differences in brain structure independent of confounding pathways. To ensure our sample is at least *in principle* sensitive to observing brain-behavior associations (cf. *Liu et al., 2023*), we reran the analysis as a simple association instead. This allows us to examine whether more years of education are associated with differences in brain structure. Notably, such an observational analysis (and resulting parameter estimate) would reflect an indeterminate mixture of causal effects as well as any indirect, sociodemographic, and individual pathways. These associations would still be of considerable potential scientific interest, but could not be interpreted as *causal* effects of education on brain structure. Crucially, this was done using the same subset of participants as in the local randomization analysis. Resulting in an estimate of how much one additional year of education correlates with brain structure.

First, we estimated the association between education (in years of attainment) and global neuroimaging measures using the same sample of participants from the one-month window local randomization analysis (i.e. August and September 1957; n~230). Similar to the causal approach, five of the six measures showed evidence in support of the null hypothesis (*Supplementary file 4*). In contrast, total SA showed weak evidence in support of a positive association of education ($BF_{10}=2.3$, n=229). To increase power, we expanded our observational analysis to participants born in a 5-month window around the ROSLA cutoff (*Figure 3*). This considerably increased the strength of evidence in support of a positive association between years of education and total SA ($BF_{10}=41.7$, N=1185). In addition, the larger pool of participants provided extreme evidence in *support* of an association between years of education and cerebral spinal fluid volume ($BF_{10}=80.7$, n=1193). This analysis highlights the stark difference between a causal and associational approach, while also providing evidence of sensitivity to brain-behavioral associations in the same sample. Of the remaining four global measures, three meaningful decreased in their strength of evidence (in terms of Jeffrey criteria *Jeffreys, 1961*) in favor of the null when compared to the causal estimate (*Figure 3*; *Supplementary file 6*). The only global neuroimaging measure that provided a similar amount of evidence (in favor of the null) between a causal and correlative approach was mean CT ($BF_{01}=7.22$ and $BF_{01}=8.81$, respectively).

A post hoc replication of this associational analysis in eight additional 10-month cohorts spaced 2 years apart (*Figure 3—figure supplement 2*) indicates our preregistered report on the associational effect of educational attainment on cerebrospinal fluid (CSF) to be most likely a false positive (*Figure 3—figure supplement 3*). Yet, the positive association between SA and educational attainment is robust across the additional eight replication cohorts.

## Discussion

In a large, preregistered study, we find converging evidence *against* a causal effect of education on long-term structural neuroimaging outcomes. This null result is present across imaging modalities, different regions, and analysis strategies. We find no issues with the design of the 1972 ROSLA, substantiating it as a valid natural experiment, in agreement with prior work (*Davies et al., 2018*; *Clark and Royer, 2010*; *Clark and Royer, 2013*; *Barcellos et al., 2023*; *Barcellos et al., 2018*). Despite a large sample (min N=4238), we find no evidence of an effect of education on any of the global neuroimaging measures with a continuity-based RD analysis. Confirming this result, we find strong evidence in support of the null hypothesis for these global neuroimaging measures using a Bayesian local randomization analysis. Moreover, we find no regionally specific effect of education on local mean CTs or SA across 66 cortical regions. This lack of localized effects was further confirmed in weighted mean fractional anisotropy for 27 white matter tracks, as well as subcortical gray matter volume in 18 regions. Moreover, we demonstrate the ability to find strong evidence in favor of observational associations between education and brain structure at this resolution, suggesting our findings are not due to lack of sensitivity more generally.

Our robust null result is seemingly at odds with causal inference findings of education's positive behavioral effect on intelligence (*Ceci, 1991*; *Judd et al., 2022*; *Gorman, 2023*) – which is sustained

throughout decades (*Ritchie and Tucker-Drob, 2018*; *Davies et al., 2018*). This juxtaposition suggests that, to the extent that the additional year of education induced long-term changes in cognitive abilities, the neural manifestation is at a level of resolution not detectable with conventional MRI field strengths. However, this would seem to contrast with a range of influential findings demonstrating that high-intensity experimental behavioral interventions (e.g. juggling, studying, memory training) lead to measurable differences in brain structure (with similar imaging pipelines) in much smaller samples (*Draganski et al., 2006*; *Draganski et al., 2004*; *Bråthen et al., 2022*). Moreover, compared to even these high-intensity interventions, a year of education is an *extensive* period of learning. The 1972 ROSLA was well implemented, schools had time to prepare and were given additional funding, increasing standardized formal qualifications of those affected (*Clark and Royer, 2013*; *Barcellos et al., 2023*). This leaves open the question of how to interpret this constellation of findings.

One potential explanation to account for this discrepancy is the concept of *expansion-renormalization* (*Wenger et al., 2017*; *Garzón et al., 2023*), which posits that following a period of skill acquisition, the cortex initially expands and then renormalizes over the course of a few months. In our context, this would suggest that the additional year would have manifested at a level detectable in MRI when the difference in educational exposure between children pre- and post-ROSLA was most pronounced and recent. In other words, MRI effects at the macro-scale might have been detectable immediately post-ROSLA in 16-year-old adolescents, before renormalizing to a micro-scale, leaving in place permanent, but microstructural changes (*Hille et al., 2024*). Possible cellular candidates for initial experience-dependent plasticity are an increase in dendritic spines, the swelling of astrocytes, and intracortical myelin adaptations (*Hille et al., 2024*; *Mediavilla et al., 2022*; *Schmidt et al., 2021*). These structural changes may be detectable using other approaches such as in vivo cellular work (cf. *Goriounova et al., 2018*), extreme high field strengths (*Garzón et al., 2023*; *Vachha and Huang, 2021*), or postmortem histology (*Perneczky et al., 2009*; *López-Otín et al., 2023*).

Additionally, the long period of time between additional education and neuroimaging offers both strengths and weaknesses for our design. First, it could be the case that 46 years is too long and any potential effect faded out over the years. That is, rather than having a micro-neural effect, it may be that there simply are no lingering effects at the brain level at all. In this case, it may be better to think of the (causal) impact of additional education as more akin to fitness or strength interventions which are also unlikely to persist across such a period. However, we note that prominent aging-related theories of heterogeneity argue *directly against* this rationale, instead positing life course experiences offer a reserve or 'brain buffer' that leads to an increasing cascade of processes limiting adverse aging effects (*Cabeza et al., 2018*; *Buckner, 2004*; *Stern, 2002*). Here, the timing between our intervention (ROSLA) and scanning makes our design particularly well-powered to test these theories, since education would contribute to an initial brain buffer (intercept) and any cumulative educational effect (slope) over 46 years. While our results are at odds with prior conceptual and observational work, a recent longitudinal study found prior education to not affect the rate of brain aging (*Nyberg et al., 2021*) – in alignment with our findings.

Lastly, to demonstrate the importance of controlling for unobserved confounders, we conducted a simple correlation analysis – using the same subset of participants – relating education to differences in brain structure. We found evidence to support an *association* between more years of education and greater SA and cerebral spinal fluid volume. This result emphasizes the need for caution in attributing causation in non-causal designs, as unobserved confounders can masquerade as an effect of interest. For instance, more education is frequently highlighted as offering behavioral and neural protection against the adverse effects of aging (*Cabeza et al., 2018*; *Buckner, 2004*; *Stern, 2002*) – while we replicate this inference associationally, we find no causal evidence for any neuroprotective effects. This suggests a more complex pathway of effects unfolding over time. Environmental causes are most likely very small and additive, which makes them not only difficult to study (*Burt and Johnson, 2023*) but also equally hard to adequately control (*von Stumm and d'Apice, 2022*). Our findings suggest that to truly understand the neural and behavioral processes that unfold after interventions such as education, we need a multipronged, mixed methods approach that combines deep phenotyping, longitudinal imaging, and behavioral follow-up (*Pfeifer et al., 2018*; *Molenaar, 2004*; *Schwartz and Glymour, 2023*), as well as more sophisticated models that can capture gene-by-environment interplay (*Judd et al., 2022*; *Burt and Johnson, 2023*; *Dolan et al., 2021*). Only then will we be able to

identify idiosyncratic environmental effects and individual characteristics underlying heterogenous lifespan development.

The UK Biobank is known to have *'healthy volunteer bias'*, as respondents tend to be healthier, more educated, and are more likely to own property (*Fry et al., 2017*; *van Alten et al., 2024*; *Lyall et al., 2022*). Various types of selection bias can occur in non-representative samples, impacting either internal (type 1) or external (type 2) validity (*van Alten et al., 2024*; *Munafò et al., 2018*). One benefit of a natural experimental design is that it protects against threats to internal validity from selection bias (*Barcellos et al., 2023*). Design-based internal validity threats still exist, such as if volunteer bias differentially impacts individuals based on the cutoff for assignment. A more pressing limitation – in particular, for an education policy change – is our power to detect effects using a sample of higher-educated individuals. This is evident in our first stage analysis examining the percentage of 15-year-olds impacted by ROSLA, which we estimate to be 10% in neuro-UKB (*Figure 1—figure supplement 2* and *Supplementary file 2*), yet has been reported to be 25% in the UK general population (*Clark and Royer, 2010*). Our results should be interpreted for this subpopulation (UK, 1973, from 15 to 16 years of age, compliers) as we estimate a 'local' average treatment effect (*Choi and Lee, 2018*). Natural experimental designs such as ours offer the potential for high internal validity at the expense of external validity.

Here, we report a lack of causal evidence of a year of school on long-term neural outcomes in a large sample of aging individuals. An additional year of education is a substantial intervention, and our preregistered findings are robust across imaging modalities, different regions, and analysis strategies. While our design cannot inform us of any short-term neural effects of education, our results call into question *sustained* experience-dependent plasticity, with significant ramifications for prominent theories of aging-related heterogeneity. The recent availability of large neuroimaging cohort data paired with cutting-edge methods from econometrics offers new avenues in studying neural effects. Causal inference is a new tool to the neuroimager's toolkit – opening novel, societally relevant phenomena – with the potential to move the field of population neuroscience from one of *association* to one of *causation*.

## Methods

On September 1, 1972 the minimum age to leave school was increased from 15 years of age to 16 in England, Scotland, and Wales (*ROSLA, 1972*). This law, henceforth ROSLA, mandated children born after September 1, 1957 to stay in school for an additional year. In contrast, a child born only one day earlier was unaffected and legally allowed to stop formal schooling at 15 years of age. The consequence of this law change was substantial: it resulted in almost 100% of children aged 15 staying in school for an additional year, in turn, increasing formal qualifications (*Figure 1—figure supplement 2*; *Clark and Royer, 2010*; *Clark and Royer, 2013*). Crucially for our purposes, ROSLA is a well-studied natural experiment with (commonly agreed, e.g. *Davies et al., 2018*; *Clark and Royer, 2010*; *Clark and Royer, 2013*; *Barcellos et al., 2023*) high design validity. A sizable body of prior work has found behavioral effects from the 1972 ROSLA (*Davies et al., 2018*; *Clark and Royer, 2010*; *Clark and Royer, 2013*; *Barcellos et al., 2023*; *Barcellos et al., 2018*).

In this study, we leverage ROSLA to study the causal effect of an additional year of education on the brain. To do so, we will use the neuroimaging sub-sample of the UK BioBank – the largest neuroimaging study to date – which also lines up perfectly with geographic and birth window (~1935–1971) requirement characteristics for ROSLA (*Littlejohns et al., 2020*). We followed our preregistration (https://osf.io/rv38z) closely, yet some minor deviations were necessary and explicitly outlined in *'deviations from preregistration'* (code: https://github.com/njudd/eduBrain, copy archived at *Judd and Kievit, 2025*).

### Structural neuroimaging outcomes

It is very plausible that a broad intervention like education could affect the brain in either a global or more regionally specific manner. For this reason, we examined both whole-brain averaged measures in addition to regional specificity in atlas-based regions and tracts. We tested the following global measures: total SA, average CT, total brain volume normalized for head size (TBV), mean wFA, WMh, and cerebral spinal fluid volume normalized for head size (CSF). Next, we examined CT and SA

regionally using the Desikan-Killiany Atlas (*Desikan et al., 2006*). The temporal pole was not included by the UK BioBank, making the total number of regions 66. We will also test weighted mean fractional anisotropy (wFA) with a global average and, regionally on 27 white matter tracks (*de Groot et al., 2013*). Lastly, we examine subcortical volume in 18 regions (*Fischl et al., 2002*).

All measures are derived from the image preprocessing pipeline from the UKB, and further preprocessing details are outlined elsewhere (*Alfaro-Almagro et al., 2018*; *McCarthy, 2020*). Outliers were winsorized if they were either above quartile 3 plus 1.5 times the interquartile range (IQR) or below quartile 1 minus 1.5 times the IQR and brought to the fence by manually recoding them to this limit (Tukey/Boxplot Method). Our alpha level for global neuroimaging measures (mean CT, total SA, average FA, WM hyper-intensities, normalized TBV and normalized CSF) is 0.05. The regional metrics (SA, CT, wFA, and subcortical structures) are false discovery rate (FDR) corrected using the number of regions per modality (e.g. 66 regions for SA) with a q value less than 0.05 considered significant. Neuroimaging measures are reported in raw units.

## Continuity-based framework: Local-linear fuzzy regression discontinuity

RD is a technique we use to estimate the effect of an intervention on an outcome where assignment (usually binary) is based on a cutoff of a running or 'forcing' variable (*Lee and Lemieux, 2010*; *Cattaneo and Titiunik, 2022*) – in our case, age in months. One major design issue in RD is if participants select into (or out of) the treatment group by sorting around to the cutoff of the running variable. However, as the 1972 ROSLA law was not pre-announced, generally strictly enforced, and affected teenagers, such alternative explanations are highly improbable (*Geruso and Royer, 2018*). Nevertheless, we conducted a density test of the running variable to check for bunching near the cutoff (*Cattaneo et al., 2018*). Although the design validity of the 1972 ROSLA is well established (*Davies et al., 2018*; *Clark and Royer, 2010*; *Clark and Royer, 2013*; *Barcellos et al., 2023*; *Barcellos et al., 2018*), we still tested a variety of placebo outcomes (outlined in 'covariates of no interest') – outcomes implausible to be affected by the intervention.

RD designs, like ours, can be 'fuzzy' indicating when assignment only increases the probability of receiving it, and in turn, treatment assigned and treatment received do not correspond for some units (*Cattaneo and Titiunik, 2022*; *Cattaneo et al., 2024*). For instance, due to cultural and historical trends, there was an increase in school attendance before ROSLA; most adolescents were continuing with education past 15 years of age (*Figure 3—figure supplement 2b*). Prior work has estimated that 25% of children would have left school a year earlier if not for ROSLA (*Clark and Royer, 2010*). Using the UK Biobank, we estimate this proportion to be around 10%, as the sample is healthier and of higher SES than the general population (*Figure 1—figure supplement 2*; ; *Fry et al., 2017*; *van Alten et al., 2024*; *Lyall et al., 2022*).

RD analysis broadly falls into two separate but complementary frameworks (*Cattaneo et al., 2019*; *Cattaneo et al., 2024*). The first, continuity-based approach defines the estimand as the difference between the limits of two continuous non-parametric functions: one fit using only participants right before the policy change to another function fit on participants right after. The other, so-called *local randomization* approaches assume participants are *'as if random'* in a small window ($\omega$) around the cutoff (described more in detail in the section 'Bayesian Local Randomization analysis'). In this case, the estimand is the mean group difference between participants before and after the cutoff within $\omega$. As $\omega$ approaches zero around the cutoff, the estimand becomes conceptually more similar to continuity-based approaches.

To empirically test whether an additional year of education caused long-lasting global and regional neural changes, we used a fuzzy local linear RD design (a continuity-based approach) with robust confidence intervals from the RDHonest package (*Lee and Lemieux, 2010*; *Armstrong and Kolesár, 2018*; *Armstrong and Kolesár, 2020*). Our outcome variables are the neuroimaging metrics described above, which were adjusted to increase statistical precision (see section 'covariates of no interest'). The running variable ($X$) is a participant's date of birth in months (mDOB). As is convention, it was centered at zero around the birth cutoff of ROSLA (September 1, 1957). Our first-stage (fuzzy) outcome was a dummy coded variable of whether the participant completed at least 16 years of education. Participants who indicated they completed college were not asked this question; therefore, we recorded their response as 21 years (*Davies et al., 2018*). Our choice of using the RDHonest package was primarily due to its ability to provide accurate inference with discrete running variables

(in our case mDOB). We used the default settings of local-linear analysis on MSE-derived bandwidths with triangular kernels (*Armstrong and Kolesár, 2018*; *Armstrong and Kolesár, 2020*).

We included participants born in England, Scotland, or Wales 10 years on either side of the September 1, 1957 cutoff (DOB September 1, 1947 – August 31st, 1967) with neuroimaging data. The range of the running variable (age in our case) to include on either side of the cutoff (known as the bandwidth; $z$) is one of the most consequential analytical decisions in RD designs. Prior work using ROSLA in the UK BioBank has analyzed bandwidths as large as 10 years (*Barcellos et al., 2018*) to as small as a year (*Davies et al., 2018*). Large bandwidths include more participants (decreasing variance), yet these participants are also further away from the cutoff, in turn, potentially increasing bias in the estimand (*Cattaneo et al., 2019*; *Cattaneo et al., 2024*). Conversely, smaller bandwidths provide less biased, yet noisier estimates. State-of-the-art continuity-based RD methods use data-driven bandwidth estimation such as mean squared error (MSE) optimized bandwidths (*Calonico et al., 2014*). Since we used this approach, the optimal bandwidth range and, in turn, the number of included participants will differ per fitted model. This also makes it logically incompatible to sensitivity test our bandwidths (*Cattaneo et al., 2019*), since they are mean squared error derived – widening them will lower variance and, in turn, increase bias while tightening them will have the opposite effect. Lastly, the use of triangular kernels means participants further away from the cutoff will be weighted less than those closer (*Cattaneo et al., 2019*). Both MSE-optimized bandwidths and triangular kernels determine the number of 'effective observations' to be fit by a fuzzy local linear RD model.

For our global continuity-based analysis, we made sure the results did not change due to missing covariate data. This was accomplished by imputing missing covariate data (≈4%) with classification and regression trees from the MICE package (*van Buuren and Groothuis-Oudshoorn, 2011*). We imputed using information from only the variables included in each analysis and did not use the running variable (DOB in months) or our first-stage instrument for prediction. We did this ten times, checking the estimates and inferences across each iteration to ensure robustness.

## Bayesian local randomization analysis

As a robustness test and to provide evidence for the null hypothesis, we conducted a Bayesian analysis using the local randomization framework for RD (*Cattaneo and Titiunik, 2022*). This alternative framework assumes a small window ($\omega$) around the cutoff (c) where the running variable is treated 'as if random' (*Cattaneo et al., 2024*). While the local randomization framework invokes stricter assumptions on the assignment mechanism, placing more importance on the window around the cutoff ($\omega$ = [c − w, c+w]), it handles discrete running variables well (*Cattaneo et al., 2024*). As the number of months on either side of the cut-off increases, bias is introduced as subjects become less similar, yet at the same time, the sample size increases, thereby lowering the variance of the estimate.

As recommended (*Cattaneo and Titiunik, 2022*; *Cattaneo et al., 2024*), we included participants within the smallest window possible ($\omega$=1 month; August vs September 1957), then expanded to a 5-month window around September 1, 1957. A dummy variable (ROSLA) was constructed to reflect if a participant was impacted by the policy change. We then tested the effect of this variable on our six global neuroimaging measures while correcting for the covariates listed above. As preregistered, in this framework we report the 'intent to treat estimate' (ITT) which should be interpreted as the effect of the policy change ROSLA on neuroimaging outcomes. Lastly, a few neuroimaging covariates did not have sufficient observations to be included (see 'deviations from preregistration').

Models were fit in R (v. 4.3.2) with rstanarm with Markov Chain Monte Carlo sampling of 80,000 iterations over four chains (*Goodrich et al., 2024*). All priors ($p$) used a normal distribution centered at 0 with autoscaling [$p$*sd(y)/sd(x)]. Our preregistration referred to using the *'default'* weakly informative prior of STAN (i.e. 2.5 SDs; *Stan Development Team, 2023*). However, this is a relatively wide prior for point null Bayesian hypothesis testing, and at odds with the defaults from packages meant for this purpose (e.g. Bayes Factor). We therefore deviated from our preregistration and reported Bayes Factors with a normal prior centered at 0 with a standard deviation of 1 (medium informative). We also report strongly informative (SD = .5) and weakly informative (SD = 1.5) normal priors (*Figure 3— figure supplement 1*).

Model diagnostics were checked with trace plots, posterior distributions, and rhat values (<1.05; *Depaoli and van de Schoot, 2017*). We then computed Bayes Factors using the bayestestR package (*Makowski et al., 2019*) using the Savage-Dickey density ratio with a point-null of 0 for each of the

3 priors. Bayes factors are reported as $BF_{10}$ in support of the alternative hypothesis. We report Bayes factors under 1 as the multiplicative inverse ($BF_{01}$=1/BF). The strength of evidence was interpreted on a graded scale using the criteria preregistered (*Jeffreys, 1961*). If the two frameworks disagree, our primary inference will be based on the continuity-based framework.

Similarly to our continuity-based approach, we conducted placebo outcome tests within both windows ($\omega$=1 and 5 months) using each included covariate as the outcome being predicted by ROSLA. In addition, we preregistered four placebo cutoffs – an analysis where the cutoff is artificially moved to test the specificity of the effect – yet null findings made this test no longer necessary (see 'deviations from preregistration').

## Correlational analysis

To compare our results to a correlational approach, we tested self-reported years of total education (EduYears) on the six global neuroimaging measures using the same participants as the local randomization analysis ($\omega$=1 and 5 months). Educational attainment is commonly used to report the effect of education on brain structure (*Judd et al., 2020*). The included subjects are identical to the local randomization approach, except the dummy coded term ROSLA was substituted for continuous EduYears.

We hypothesize that the *associational* effect of educational attainment on global neuroimaging measures is the result of various unmeasured individual and societal-level factors confounding this relationship. Upon a reviewer-initiated comment (see Elife Reviewer 1 Recommendation 7), we made eight more 10-month sub-cohorts spaced two years apart (4 after ROSLA and before ROSLA; *Figure 3—figure supplement 2*). In each of these eight cohorts, we tested the effect of educational attainment on our six global neuroimaging covariates following identical methodology Bayesian model, normal prior (mean = 0, sd = 1). From these models, we extracted Bayesian posterior estimates of the associational effect of educational attainment and plotted them (*Figure 3—figure supplement 3*).

## Covariates of no interest

In neuroimaging, it is common to include covariates. However, for identification in a valid RD design, covariates *by definition* should be equal on either side of the cutoff (*Cattaneo et al., 2019*). We used standard neuroimaging covariates for two purposes: (*1*) to further test the validity of the 1972 ROSLA and (*2*) to increase statistical precision in our RD analysis. For instance, there are large sex differences in certain neural measures mostly related to head size (*Ritchie et al., 2018*); therefore, we included a dummy variable for sex. We expect ROSLA to not affect the proportion of males and females, yet including this measure as a covariate will increase the estimation precision for measures sensitive to sex-related head size differences (e.g. SA [as expected, we found a 24% precision gain for total SA, measured as the difference in CI width of the RD parameter pre and post-correction. Precision increased for *all* global measures (3–7%, 24%).]).

Children born in July and August could technically leave school at 15 after the 1972 ROSLA due to the exam period ending in mid-June and the school year starting on September 1 (*Clark and Royer, 2010*). This is an artifact of asking in the UK Biobank *'What age did you complete full-time education?'*, rather than *'How many years of education did you complete?'*. This artifact predates and is not influenced by ROSLA (*Avendano et al., 2020*). Yet to account for this, we included a dummy variable for children born in July or August.

Lastly, Neuro-UKB recruitment is still ongoing and started in April 2014 (*Littlejohns et al., 2020*). In turn, there is a wide range of intervals in terms of the period between when someone was born and the age at which they were scanned (*Figure 1—figure supplement 1*). We control for this discrepancy by including a variable 'date of scanning' (DoS). This was done by coding the earliest date a (included) participant was scanned to 0 and counting the number of days until the last included participant. This resulted in a value for each participant that was the number of days they were scanned after the first participant. We also include this variable squared, so we can model quadratic effects, resulting in two variables (DoS and DoS *UNCRC, 1989*). Other (preregistered) neuroimaging quality control measures were included, such as scanning site, head motion, whether a T2 FLAIR sequence was used, T1 intensity scaling, and three diffusion measures recommended (*Smith et al., 2024*). No additional covariates other than those preregistered (https://osf.io/rv38z) were added.

Our first aim, known as a placebo outcome test, involved testing the effect of an additional year of education on each covariate. This was accomplished identically to our neuroimaging outcomes (see section 'Continuity-based framework') using RDHonest. We expect there to be no effect of ROSLA on any covariate (*Supplementary files 5 and 7*). This was also tested (and confirmed) for the Local-Randomization Framework ($\omega$=1 and 5 months).

For our second aim, increasing the precision of the RD estimand, we used covariate-adjusted outcomes (Y). This was done by fitting a local linear regression for each MSE-derived bandwidth $z$ with a matrix of covariates (C') on the unadjusted outcome (uY) using *Equation 1* below. Let X denote the running variable (DOB in months) which is centered around the ROSLA cutoff (i.e. September 1, 1957=0). In the second step, this matrix of covariates is then multiplied by the fitted coefficients ($\hat{\beta}_*$) and subtracted from the unadjusted outcome (uY) to make a covariate-adjusted outcome (Y). Since our assignment mechanism is probabilistic (i.e. fuzzy RD), we also corrected our first-stage outcome, a dummy coded variable reflecting if the participant stayed in school until 16, in an identical manner. This method was used after personal communication with Prof. Michal Kolesar, the package creator of RDHonest. The latest version of the package includes covariate correction, and this is preferred to our method outlined below. For the Bayesian local randomization analysis and correlational analysis, we simply included the covariates in the model.

$$uY = \beta_0 + \beta_1 \left( X \right) + \beta_2 \left( X \geq 0 \right) + \beta_3 \left( X * X \geq 0 \right) + \beta_* C' + \epsilon$$
$$Y = uY - \left( \hat{\beta}_* \times C' \right)$$

## Deviations from the preregistration

The study closely followed the preregistration pipeline (https://osf.io/rv38z), yet in a few minor cases, it was not possible to follow. For instance, our initial plan involved placebo cutoffs – a falsification test where the cutoff is artificially moved to check if a significant result may also appear (for whatever reason) at other, non-hypothesized dates. Due to our lack of findings, we did not conduct this analysis as it was not necessary.

Some covariates lacked variance in specific specifications due to very few observations and therefore could not be included. In the global continuity analysis, this only impacted wFA, where the variable 'T2_FLAIR' was not used. This dummy-coded variable, indicating if a participant had a T2-weighted MRI scan, was also not used in the regional continuity-based analysis nor for the local-randomization analysis. The local randomization analysis included a small number of participants around September 1, 1957; therefore, we did not include the covariate 'summer' as this would have been isomorphic to our effect of interest (ROSLA). Lastly, for the 1-month window analysis, there were not enough observations to include imaging center 11028.

## Choice of priors

The Savage-Dickey density ratio is the height of the posterior divided by the height of the prior at a particular point (0 in our case for Point null Bayes Factors). This makes them particularly sensitive to the prior used. Our preregistration referred to using a '*default weakly informative prior*'. While not specified, this was referencing the default prior of STAN (i.e. normal of 2.5 SDs). However, this is arguably too wide for adequate point null Bayesian hypothesis testing and at odds with the defaults from packages meant for this purpose (e.g. BayesFactor package). If we used the default prior from STAN, it would have given us unrealistically strong support for the null hypothesis. We therefore deviated from our preregistration and reported Bayes Factors with a normal prior centered at 0 with a standard deviation of 1 (mediumly informative). We also report strongly informative (SD = 0.5) and weakly informative (SD = 1.5) normal priors both also centered at 0 (*Figure 3—figure supplement 1*). All of our priors supported the null hypothesis; they just varied in the amount of evidence in support of the null. Lastly, for illustration purposes, we ran the 5-month local randomization analysis for SA regions using the mediumly informative prior (*Figure 2*).

## Post hoc associational replication

In our associational analysis, we used educational attainment in years in the identical subset of subjects used for the local randomization causal analysis. This allowed us to compare and contrast the weight of evidence in a causal and associational analysis (*Figure 3*) within the same subjects. Upon a

reviewer-initiated comment (see Elife v1 Reviewer 1 Recommendation 7), we made eight additional subcohorts (see *Figure 3—figure supplement 2*) spaced 2 years apart, as the placement of our associational analysis should not matter. We found our results to hold in all global neuroimaging covariates bar CSF, which seemed to only show an effect in our specific preregistered cohort (*Figure 3—figure supplement 3*).

## Acknowledgements

Nicholas Judd is supported with eScience, Jacobs, and Pro Futura Scientia fellowships by the Netherlands eScience institute, Jacobs foundation, and Riksbankens Jubileumsfond, respectively. Rogier Kievit is supported by a Hypatia fellowship from the RadboudUMC. This research has been conducted using the UK Biobank resource under application number 23668. We would like to thank Barbara Sakic, Barbara Franke, Andre Marquand, Jan Buitelaar, Nina Roth Mota, Janita Bralten, and Ward de Witte for help in navigating and accessing the UK Biobank. We would like to thank Prof. Michal Kolesár for his help in covariate correction, along with highlighting the great code base from Dr. Margot van de Weijer (*van de Weijer, 2022*).

## Additional information

### Funding

| Funder | Grant reference number | Author |
| --- | --- | --- |
| Jacobs Foundation | Jacobs Fellowship | Nicholas Judd |
| Radboud Universitair Medisch Centrum | Hypatia fellowship | Rogier Kievit |
| Riksbankens Jubileumsfond | Pro Futura Fellowship | Nicholas Judd |

The funders had no role in study design, data collection and interpretation, or the decision to submit the work for publication.

### Author contributions

Nicholas Judd, Conceptualization, Data curation, Formal analysis, Visualization, Methodology, Writing – original draft, Writing – review and editing; Rogier Kievit, Conceptualization, Funding acquisition, Writing – original draft, Project administration, Writing – review and editing

### Author ORCIDs

Nicholas Judd ⓘD https://orcid.org/0000-0002-0196-9871
Rogier Kievit ⓘD https://orcid.org/0000-0003-0700-4568

### Ethics

Ethical StatementUK Biobank has ethical approval from the North West Multi-centre Research Ethics Committee (MREC) as a Research Tissue Bank (RTB) (approval number: 11/NW/0382). This consortium has its own independent ethics advisory committee (https://www.ukbiobank.ac.uk/learn-more-about-uk-biobank/governance/ethics-advisory-committee), that assures the UK Biobank abiders by the ethics and governance framework (https://www.ukbiobank.ac.uk/media/0xsbmfmw/egf.pdf). Written informed consent was obtained from all participants.

Reviewer #2 (Public review): https://doi.org/10.7554/eLife.101526.3.sa1
Reviewer #3 (Public review): https://doi.org/10.7554/eLife.101526.3.sa2
Author response https://doi.org/10.7554/eLife.101526.3.sa3

## Additional files

### Supplementary files
Supplementary file 1. Fuzzy RD Placebo Outcome results.

Supplementary file 2. Fuzzy RD Global Neuroimaging Results.

Supplementary file 3. Fuzzy RD Uncorrected Global Neuroimaging Results.

Supplementary file 4. One month window Bayesian Analysis.

Supplementary file 5. One Month window Test of Covariates.

Supplementary file 6. Five Month window Bayesian Analysis.

Supplementary file 7. Five Month window Test of Covariates.

MDAR checklist

### Data availability
All code is publicly available (https://github.com/njudd/eduBrain, copy archived at *Judd and Kievit, 2025*) for a very similar more streamlined script see (https://github.com/njudd/EduTelomere, copy archived at *Judd, 2024*). The data is also publicly available yet must be accessed via the centralized UK BioBank repository (https://www.ukbiobank.ac.uk). This research has been conducted using the UK Biobank resource under application number 23668.

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
