## [Editor Report · eLife Assessment]

A regression discontinuity analysis finds essentially no effect of 1 additional year of secondary education on brain structure in adulthood. This is a **valuable** finding that adds to the literature on the impact of education on brain health. While the finding is **convincing** on its own, as the analysis was pre-registered and very carefully conducted, the impact is limited as the manipulated variable only relates to a single additional year of education (remaining in education to 15 vs 16 years of age).

---

## [Referee Report · Reviewer #2 (Public review)]

Summary:

The authors conduct a causal analysis of years of secondary education on brain structure in late life. They use a regression discontinuity anlaysis to measure the impact of a UK law change in 1972 that increased the years of mandatory education by 1 year. Using brain imaging data from the UK Biobank, they find essentially no evidence for 1 additional year of education altering brain structure in adulthood.

Strengths:

The authors pre-registered the study and the regression discontinuity was very carefully described and conducted. They completed a large number of diagnostic and alternate analyses to allow for different possible features in the data. (Unlike a positive finding, a negative finding is only bolstered by additional alternative anlayses).

Weaknesses:

While the work is of high quality for the precise question asked, ultimately the exposure (1 additional year of education) is a very modest manipulation and the outcome measured long after the intervention. Thus a null finding here is completely consistent educational attainement (EA) in fact having an impact on brain structure, where EA may reflect elements of training after second education (e.g. university, post-graduate qualifications, etc) and not just stopping education at 16 yrs yes/no.

---

## [Referee Report · Reviewer #3 (Public review)]

Summary:

This study investigates evidence for a hypothesised, causal relationship between education, specifically the number of years spent in school, and brain structure as measured by common brain phenotypes such as surface area, cortical thickness, total volume and diffusivity.

To test their hypothesis, the authors rely on a "natural" intervention, that is, the 1972 ROSLA act that mandated an extra year of education for all 15-year olds. The study's aim is to determine potential discontinuities in the outcomes of interest at the time of the policy change, which would indicate a causal dependence. Naturalistic experiments of this kind are akin to randomised controlled trials, the gold standard for answering questions of causality.

Using two complementary, regression-based approaches, the authors find no discernible effect of spending an extra year in primary education on brain structure. The authors further demonstrate that observational studies showing an effect between education and brain structure may be confounded and thus unreliable when assessing causal relationships.

Strengths:

- A clear strength of this study is the large sample size totalling up to 30k participants from the UK Biobank. Although sample sizes for individual analyses are an order of magnitude smaller, most neuroimaging studies usually have to rely on much smaller samples.

- This study has been preregistered in advance, detailing the authors' scientific question, planned method of inquiry and intended analyses, with only minor, justifiable changes in the final analysis.

- The analyses look at both global and local brain measures used as outcomes, thereby assessing a diverse range of brain phenotypes that could be implicated in a causal relationship with a person's level of education.

- The authors use multiple methodological approaches, including validation and sensitivity analyses, to investigate the robustness of their findings and, in the case of correlational analysis, highlight differences with related work by others.

- The extensive discussion of findings and how they relate to the existing, somewhat contradictory literature gives a comprehensive overview of the current state of research in this area.

Weaknesses:

- This study investigates a well-posed but necessarily narrow question in a specific setting: 15-year old British students born around 1957 who also participate in the UKB imaging study roughly 60 years later. Thus conclusions about the existence or absence of any general effect of the number of years of education on the brain's structure are limited to this specific scenario.

- The modelling approach used in this study requires that all covariates of no interest are equal before and after the cut-off, something that is impossible to test. However, other studies have not found specific issues that would invalidate ROSLA as a natural experiment.

---

## [Author Response]

The following is the authors’ response to the original reviews

**Reviewer #1 (Public review):**
Summary:This fascinating manuscript studies the effect of education on brain structure through a natural experiment. Leveraging the UK BioBank, these authors study the causal effect of education using causal inference methodology that focuses on legislation for an additional mandatory year of education in a regression discontinuity design.Strengths:The methodological novelty and study design were viewed as strong, as was the import of the question under study. The evidence presented is solid. The work will be of broad interest to neuroscientistsWeaknesses:There were several areas which might be strengthed from additional consideration from a methodological perspective.

We sincerely thank the reviewer for the useful input, in particular, their recommendation to clarify RD and for catching some minor errors in the methods (such as taking the log of the Bayes factors).

**Reviewer #1 (Recommendations for the authors):**
(1) The fuzzy local-linear regression discontinuity analysis would benefit from further description.(2) In the description of the model, the terms "smoothness" and "continuity" appear to be used interchangeably. This should be adjusted to conform to mathematical definitions.

We have now added to our explanations of continuity regression discontinuity. In particular, we now explain “fuzzy”, and add emphasis on the two separate empirical approaches (continuity and local-randomization), along with fixing our use of “smoothness” and “continuity”.

results:

“Compliance with ROSLA was very high (near 100%; Sup. Figure 2). However, given the cultural and historical trends leading to an increase in school attendance before ROSLA, most adolescents were continuing with education past 15 years of age before the policy change (Sup Plot. 7b). Prior work has estimated 25 percent of children would have left school a year earlier if not for ROSLA 41. Using the UK Biobank, we estimate this proportion to be around 10%, as the sample is healthier and of higher SES than the general population (Sup. Figure 2; Sup. Table 2) 46–48.”

methods:

“RD designs, like ours, can be ‘fuzzy’ indicating when assignment only increases the probability of receiving it, in turn, treatment assigned and treatment received do not correspond for some units 33,53. For instance, due to cultural and historical trends, there was an increase in school attendance before ROSLA; most adolescents were continuing with education past 15 years of age (Sup Plot. 7b). Prior work has estimated that 25 percent of children would have left school a year earlier if not for ROSLA 41. Using the UK Biobank, we estimate this proportion to be around 10%, as the sample is healthier and of higher SES than the general population (Sup. Figure 2; Sup. Table 2) 46–48.”

(3) The optimization of the smoother based on MSE would benefit from more explanation and consideration. How was the flexibility of the model taken into account in testing? Were there any concerns about post-selection inference? A sensitivity analysis across bandwidths is also necessary. Based on the model fit in Figure 1, results from a linear model should also be compared.

It is common in the RD literature to illustrate plots with higher-order polynomial fits while inference is based on linear (or at most quadratic) models (Cattaneo, Idrobo & Titiunik, 2019). We agree that this field-specific practice can be confusing to readers. Therefore, we have redone Figure 1 using local-linear fits better aligning with our analysis pipeline. Yet, it is still not a *one-to-one* alignment as point estimation and confidence are handled robustly while our plotting tools are simple linear fits. In addition, we updated Sup. Fig 3 and moved 3rd-order polynomial RD plots to Sup. Fig 4.

Empirical RD has many branching analytical decisions (bandwidth, polynomial order, kernel) which can have large effects on the outcome. Fortunately, RD methodology is starting to become more standardized (Catteneo & Titiunik, 2022, Ann. Econ Rev) as there have been indications of publication bias using these methods (Stommes, Aronow & Sävje, 2023, Research and Politics (This paper suggest it is not researcher degrees of freedom, rather inappropriate inferential methods)). While not necessarily ill-intended, researcher degrees of freedom and analytic flexibility are major contributors to publication bias. We (self) limited our analytic flexibility by using pre-registration (https://osf.io/rv38z).

One of the most consequential analytic decisions in RD is the bandwidth size as there is no established practice, they are context-specific and can be highly influential on the results. The choice of bandwidths can be framed as a ‘bias vs. variance trade-off’. As bandwidths increase, variance decreases since more subjects are added yet bias (misspecification error/smoothing bias) also increases (as these subjects are further away and less similar). In our case, our assignment (running/forcing) variable is ‘date of birth in months’; therefore our smallest comparison would be individuals born in August 1957 (unaffected/no treatment) vs September 1957 (affected/treated). This comparison has the least bias (subjects are the most similar to each other), yet it comes at the expense of *very few* subjects (high variance in our estimate).

MSE-derived bandwidths attempt to solve this issue by offering an automatic method to choose an analysis bandwidth in RD. Specifically, this aims to minimize the MSE of the local polynomial RD point estimator – effectively choosing a bandwidth by balancing the ‘bias vs. variance trade-off’ (explained in detail 4.4.2 Cattaneo et al., 2019 p 45 - 51 “A practical introduction to regression discontinuity designs: foundations”). Yet, you are very correct in highlighting potential overfitting issues as they are “by construction invalid for inference” (Calonico, Cattaneo & Farrell, 2020, p. 192). Quoting from Cattaneo and Titiunik’s Annual Review of Economics from 2022:

“Ignoring the misspecification bias can lead to substantial overrejection of the null hypothesis of no treatment effect. For example, back-of-the-envelop calculations show that a nominal 95% confidence interval would have an empirical coverage of about 80%.”

Fortunately, modern RD analysis packages (such as rdrohust or RDHonest) calculate robust confidence intervals - for more details see Armstrong and Kolesar (2020). For a summary on MSE-bandwidths see the section “Why is it hard to estimate RD effects?” in Stommes and colleagues 2023 (https://arxiv.org/abs/2109.14526). For more in-depth handling see the Catteneo, Idrobo, and Titiunik primer (https://arxiv.org/abs/1911.09511).

Lastly, with MSE-derived bandwidths sensitivity tests only make sense within a narrow window of the MSE-optimized bandwidth (5.5 Cattaneo et al., 2019 p 106 - 107). When a significant effect occurs, placebo cutoffs (artificially moving the cutoff) and donut-hole analysis are great sensitivity tests. Instead of testing our bandwidths, we decided to use an alternate RD framework (local randomization) in which we compare 1-month and 5-month windows. Across all analysis strategies, MRI modalities, and brain regions, we do not find any effects of the education policy change ROSLA on long-term neural outcomes.

(4) In the Bayesian analysis, the authors deviated from their preregistered analytic plan. This whole section is a bit confusing in its current form - for example, point masses are not wide but rather narrow. Bayes factors are usually estimated; it is unclear how or why a prior was specified. What exactly is being modeled using a prior? Also, throughout - If the log was taken, as the methods seem to indicate for the Bayes factor, this should be mentioned in figures and reported estimates.

First, we would like to thank you for spotting that we incorrectly kept the log in the methods. We have fixed this and added the following sentence to the methods:

“Bayes factors are reported as BF_10_ in support of the alternative hypothesis, we report Bayes factors under 1 as the multiplicative inverse (BF_01_ = 1/BF)”

All Bayesian analyses need to have a prior. In practice, this becomes an issue when you’re uncertain about (1) the location of the effect (directionality & center mass, defined by a location parameter), yet more importantly, the (2) confidence/certainty of the range-spread of possible effects (determined by a scale parameter). In normally distributed priors these two ‘beliefs’ are represented with a mean and a standard deviation (the latter impacts your confidence/certainty on the range of plausible parameter space).

Supplementary figure 6 illustrates several distributions (location = 0 for all) with varying scale parameters; when used as Bayesian priors this indicates differing levels of confidence in our certainty of the plausible parameter space. We illustrate our three reported, normally distributed priors centered at zero in blue with their differing scale parameters (sd = .5, 1 & 1.5).

All of these five prior distributions have the same location parameter (i.e., 0) yet varying differences in the scale parameter – our confidence in the certainty of the plausible parameter space. At first glance it might seem like a flat/uniform prior (not represented) is a good idea – yet, this would put equal weight on the possibility of every estimate thereby giving the same probability mass to implausible values as plausible ones. A uniform prior would, for instance, encode the hypothesis that education causing a 1% increase in brain volume is just as plausible as it causing either a doubling or halving in brain volume. In human research, we roughly know a range of reasonable effect sizes and it is rare to see massive effects.

A benefit of ‘weakly-informative’ priors is that they limit the range of plausible parameter values. The default prior in STAN (a popular Bayesian estimation program; https://mc-stan.org) is a normally distributed prior with a mean of zero and an SD of 2.5 (seen in orange in the figure; our initial preregistered prior). This large standard deviation easily permits positive and negative estimates putting minimal emphasis on zero. Contrast this to BayesFactor package’s (Morey R, Rouder J, 2023) default “wide” prior which is the Cauchy distribution (0, .7) illustrated in magenta (for more on the Cauchy see: https://distribution-explorer.github.io/continuous/cauchy.html).

These different defaults reflect differing Bayesian philosophical schools (‘estimate parameters’ vs ‘quantify evidence’ camps); if your goal is to accurately estimate a parameter it would be odd to have a strong null prior, yet (in our opinion) when estimating point-null BF’s a wide default prior gives far too much evidence in support of the null. In point-null BF testing the Savage-Dickey density ratio is the ratio between the height of the prior at 0 and the height of the posterior at zero (see Figure under section “testing against point null 0”). This means BFs can be very prior sensitive (seen in SI tables 5 & 6). For this reason, we thought it made sense to do prior sensitivity testing, to ensure our conclusions in favor of the null were not caused solely by an overly wide prior (preregistered orange distribution) we decided to report the 3 narrower priors (blue ones).

Alternative Bayesian null hypotheses testing methods such as using Bayes Factors to test against a null region and ‘region of practical equivalence testing’ are less prior sensitive, yet both methods demand the researcher (e.g. ‘us’) to decide on a minimal effect size of practical interest. Once a minimal effect size of interest is determined any effect within this boundary is taken as evidence in support of the null hypothesis.

(5) It is unclear why a different method was employed for the August / September data analysis compared to the full-time series.

We used a local-randomization RD framework, an entirely different empirical framework than continuity methods (resulting in a different estimate). For an overview see the primer by Cattaneo, Idrobo & Titiunik 2023 (“A Practical Introduction to Regression Discontinuity Designs: Extensions”; https://arxiv.org/abs/2301.08958).

A local randomization framework is optimal when the running variable is discrete (as in our case with DOB in months) (Cattaneo, Idrobo & Titiunik 2023). It makes stronger assumptions on exchangeability therefore a very narrow window around the cutoff needs to be used. See Figure 2.1 and 2.2 (in the Cattaneo, Idrobo & Titiunik 2023) for graphical illustrations of (1) a randomized experiment, (2) a continuity RD design, and (3) local-randomization RD. Using the full-time series in a local randomization analysis is not recommended as there is no control for differences between individuals as we move further away from the cutoff – making the estimated parameter highly endogenous.

We understand how it is confusing to have both a new framework and Bayesian methods (we could have chosen a fully frequentist approach) but using a different framework allows us to weigh up the aforementioned ‘bias vs variance tradeoff’ while Bayesian methods allow us to say something about the weight of evidence (for or against) our hypothesis.

(6) Figure 1 - why not use model fits from those employed for hypothesis testing?

This is a great suggestion (ties into #3), we have now redone Figure 1.

(7) The section on "correlational effect" might also benefit from additional analyses and clarifications. Indeed, the data come from the same randomized experiment for which minimum education requirements were adjusted. Was the only difference that the number of years of education was studied as opposed to the cohort? If so, would the results of this analysis be similar in another subsample of the UK Biobank for which there was no change in policy?

We have clarified the methods section for the correlational/associational effect. This was the same subset of individuals for the local randomization analysis; all we did was change the independent variable from an exogenous dummy-coded ROSLA term (where half of the sample had the natural experiment) to a continuous (endogenous) educational attainment IV.

In principle, the results from the associational analysis should be exactly the same if we use other UK Biobank cohorts. To see if the association of education attainment with the global neuroimaging cohorts was similar across sub-cohorts of new individuals, we conducted post hoc Bayesian analysis on eight more subcohort of 10-month intervals, spaced 2 years apart from each other (Sup. Figure 7; each indicated by a different color). Four of these sub-cohorts predate ROSLA, while the other four are after ROSLA. Educational attainment is slowly increasing across the cohorts of individuals born from 1949 until 1965; intriguingly the effect of ROSLA is visually evident in the distributions of educational attainment (Sup. Figure 7). Also, as seen in the cohorts predating ROSLA more and more individuals were (already) choosing to stay in education past 15 years of age (see cohort 1949 vs 1955 in Sup. Figure 7).

Sup. Figure 8 illustrates boxplots of the educational attainment posterior of the eight sub-cohorts in addition to our original analysis (s1957) using a normal distributed prior with a mean of 0 and a sd of 1. Total surface area shows a remarkably replicable association with education attainment. Yet, it is evident the “extremely strong” association we found for CSF was a statistical fluke – as the posterior of other cohorts (bar our initial test) crosses zero. The conclusions for the other global neuroimaging covariates where we concluded ‘no associational effect’ seems to hold across cohorts.

We have now added methods, deviation from preregistration, and the following excerpt to the results:

“A post hoc replication of this associational analysis in eight additional 10-month cohorts spaced two years apart (Sup. Figure 7) indicates our preregistered report on the associational effect of educational attainment on CSF to be most likely a false-positive (Sup. Figure 8). Yet, the positive association between surface area and educational attainment is robust across the additional eight replication cohorts.”

**Reviewer #2 (Public review):**
Summary:The authors conduct a causal analysis of years of secondary education on brain structure in late life. They use a regression discontinuity analysis to measure the impact of a UK law change in 1972 that increased the years of mandatory education by 1 year. Using brain imaging data from the UK Biobank, they find essentially no evidence for 1 additional year of education altering brain structure in adulthood.Strengths:The authors pre-registered the study and the regression discontinuity was very carefully described and conducted. They completed a large number of diagnostic and alternate analyses to allow for different possible features in the data. (Unlike a positive finding, a negative finding is only bolstered by additional alternative analyses).Weaknesses:While the work is of high quality for the precise question asked, ultimately the exposure (1 additional year of education) is a very modest manipulation and the outcome is measured long after the intervention. Thus a null finding here is completely consistent educational attainment (EA) in fact having an impact on brain structure, where EA may reflect elements of training after a second education (e.g. university, post-graduate qualifications, etc) and not just stopping education at 16 yrs yes/no.The work also does not address the impact of the UK Biobank's well-known healthy volunteer bias (Fry et al., 2017) which is yet further magnified in the imaging extension study (Littlejohns et al., 2020). Under-representation of people with low EA will dilute the effects of EA and impact the interpretation of these results.References:Fry, A., Littlejohns, T. J., Sudlow, C., Doherty, N., Adamska, L., Sprosen, T., Collins, R., & Allen, N. E. (2017). Comparison of Sociodemographic and Health-Related Characteristics of UK Biobank Participants With Those of the General Population. American Journal of Epidemiology, 186(9), 1026-1034. https://doi.org/10.1093/aje/kwx246Littlejohns, T. J., Holliday, J., Gibson, L. M., Garratt, S., Oesingmann, N., Alfaro-Almagro, F., Bell, J. D., Boultwood, C., Collins, R., Conroy, M. C., Crabtree, N., Doherty, N., Frangi, A. F., Harvey, N. C., Leeson, P., Miller, K. L., Neubauer, S., Petersen, S. E., Sellors, J., ... Allen, N. E. (2020). The UK Biobank imaging enhancement of 100,000 participants: rationale, data collection, management and future directions. Nature Communications, 11(1), 2624. https://doi.org/10.1038/s41467-020-15948-9

We thank the reviewer for the positive comments and constructive feedback, in particular, their emphasis on volunteer bias in UKB (similar points were mentioned by Reviewer 3). We have now addressed these limitations with the following passage in the discussion:

“The UK Biobank is known to have ‘healthy volunteer bias’, as respondents tend to be healthier, more educated, and are more likely to own assets [71,72]. Various types of selection bias can occur in non-representative samples, impacting either internal (type 1) or external (type 2) validity. One benefit of a natural experimental design is that it protects against threats to internal validity from selection bias [43], design-based internal validity threats still exist, such as if volunteer bias differentially impacts individuals based on the cutoff for assignment. A more pressing limitation – in particular, for an education policy change – is our power to detect effects using a sample of higher-educated individuals. This is evident in our first stage analysis examining the percentage of 15-year-olds impacted by ROSLA, which we estimate to be 10% in neuro-UKB (Sup. Figure 2 & Sup. Table 2), yet has been reported to be 25% in the UK general population [41]. Our results should be interpreted for this subpopulation (UK, 1973, from 15 to 16 years of age, compliers) as we estimate a ‘local’ average treatment effect [73]. Natural experimental designs such as ours offer the potential for high internal validity at the expense of external validity.”

We also highlighted it both in the results and methods.

We appreciate that one year of education may seem modest compared to the entire educational trajectory, but as an intervention, we disagree that one year of education is ‘a very modest manipulation’. It is arguably one of the largest positive manipulations in childhood development we can administer. If we were to translate a year of education into the language of a (cognitive) intervention, it is clear that the manipulation, at least in terms of hours, days, and weeks, is substantial. Prior work on structural plasticity (e.g., motor, spatial & cognitive training) has involved substantially more limited manipulations in time, intensity, and extent. There is even (limited) evidence of localized persistent long-term structural changes (Wollett & Maguire, 2011, Cur. Bio.).

We have now also highlighted the limited generalizability of our findings since we estimate a ‘local’ average treatment effect. It is possible higher education (college, university, vocational schools, etc.) could impact brain structure, yet we see no theoretical reason why it would while secondary wouldn’t. Moreover, higher education education is even trickier to research empirically due to heightened self and administrative selection pressures. While we cannot discount this possibility, the impacts of endogenous factors such as genetics and socioeconomic status are most likely heightened. That being said, higher education offers exciting possibilities to compare more domain-specific processes (e.g., by comparing a philosophy student to a mathematics student). Causality could be tested in European systems with point entry into *field-specific* programs – allowing comparison of students who just missed entry criteria into one topic and settled for another.

Regarding the amount of time following the manipulation, as we highlight in our discussion this is both a weakness and a strength. Viewed from a developmental neuroplasticity lens it would have been nice to have imaging immediately following the manipulation. Yet, from an aging perspective, our design has *increased power* to detect an effect.

**Reviewer #2 (Recommendations for the authors):**
(1) The authors assert there is no strong causal evidence for EA on brain structure. This overlooks work from Mendielian Randomisation, e.g. this careful work: https://pubmed.ncbi.nlm.nih.gov/36310536/ ... evidence from (good quality) MR studies should be considered.

We thank the reviewer for highlighting this well-done mendelian randomization study. We have now added this citation and removed previous claims on the “lack of causal evidence existing”. We refrain from discussing Mendelian randomization, as it it would need to be accompanied by a nuanced discussion on the strong limitations regarding EduYears-PGS in Mendelian randomization designs.

(2) Tukey/Boxplot is a good name for your identification of outliers but your treatment of outliers has a well-recognized name that is missing: Windsorisation. Please add this term to your description to help the reader more quickly understand what was done.

Thanks, we have now added the term winsorized.

(3) Nowhere is it plainly stated that "fuzzy" means that you allow for imperfect compliance with the exposure, i.e. some children born before the cut-off stayed in school until 16, and some born after the cut-off left school before 16. For those unfamiliar with RD it would be very helpful to explain this at or near the first reference of the term "fuzzy".

We have now clarified the term ‘fuzzy’ to the results and methods:

methods:

“RD designs, like ours, can be ‘fuzzy’ indicating when assignment only increases the probability of receiving it, in turn, treatment assigned and treatment received do not correspond for some units 33,53. For instance, due to cultural and historical trends, there was an increase in school attendance before ROSLA; most adolescents were continuing with education past 15 years of age (Sup Plot. 7b). Prior work has estimated that 25 percent of children would have left school a year earlier if not for ROSLA 41. Using the UK Biobank, we estimate this proportion to be around 10%, as the sample is healthier and of higher SES than the general population (Sup. Figure 2; Sup. Table 2) 46–48.”

(4) Supplementary Figure 2 never states what the percentage actually measures. What exactly does each dot represent? Is it based on UK Biobank subjects with a given birth month? If so clarify.

Fixed!

**Reviewer #3 (Public review):**
Summary:This study investigates evidence for a hypothesized, causal relationship between education, specifically the number of years spent in school, and brain structure as measured by common brain phenotypes such as surface area, cortical thickness, total volume, and diffusivity.To test their hypothesis, the authors rely on a "natural" intervention, that is, the 1972 ROSLA act that mandated an extra year of education for all 15-year-olds. The study's aim is to determine potential discontinuities in the outcomes of interest at the time of the policy change, which would indicate a causal dependence. Naturalistic experiments of this kind are akin to randomised controlled trials, the gold standard for answering questions of causality.Using two complementary, regression-based approaches, the authors find no discernible effect of spending an extra year in primary education on brain structure. The authors further demonstrate that observational studies showing an effect between education and brain structure may be confounded and thus unreliable when assessing causal relationships.Strengths:(1) A clear strength of this study is the large sample size totalling up to 30k participants from the UK Biobank. Although sample sizes for individual analyses are an order of magnitude smaller, most neuroimaging studies usually have to rely on much smaller samples.(2) This study has been preregistered in advance, detailing the authors' scientific question, planned method of inquiry, and intended analyses, with only minor, justifiable changes in the final analysis.(3) The analyses look at both global and local brain measures used as outcomes, thereby assessing a diverse range of brain phenotypes that could be implicated in a causal relationship with a person's level of education.(4) The authors use multiple methodological approaches, including validation and sensitivity analyses, to investigate the robustness of their findings and, in the case of correlational analysis, highlight differences with related work by others.(5) The extensive discussion of findings and how they relate to the existing, somewhat contradictory literature gives a comprehensive overview of the current state of research in this area.Weaknesses:(1) This study investigates a well-posed but necessarily narrow question in a specific setting: 15-year-old British students born around 1957 who also participated in the UKB imaging study roughly 60 years later. Thus conclusions about the existence or absence of any general effect of the number of years of education on the brain's structure are limited to this specific scenario.(2) The authors address potential concerns about the validity of modelling assumptions and the sensitivity of the regression discontinuity design approach. However, the possibility of selection and cohort bias remains and is not discussed clearly in the paper. Other studies (e.g. Davies et al 2018, https://www.nature.com/articles/s41562-017-0279-y) have used the same policy intervention to study other health-related outcomes and have established ROSLA as a valid naturalistic experiment. Still, quoting Davies et al. (2018), "This assumes that the participants who reported leaving school at 15 years of age are a representative sample of the sub-population who left at 15 years of age. If this assumption does not hold, for example, if the sampled participants who left school at 15 years of age were healthier than those in the population, then the estimates could underestimate the differences between the groups.". Recent studies (Tyrrell 2021, Pirastu 2021) have shown that UK Biobank participants are on average healthier than the general population. Moreover, the imaging sub-group has an even stronger "healthy" bias (Lyall 2022).(3) The modelling approach used in this study requires that all covariates of no interest are equal before and after the cut-off, something that is impossible to test. Mentioned only briefly, the inclusion and exclusion of covariates in the model are not discussed in detail. Standard imaging confounds such as head motion and scanning site have been included but other factors (e.g. physical exercise, smoking, socioeconomic status, genetics, alcohol consumption, etc.) may also play a role.

We thank the reviewer for their numerous positive comments and have now attempted to address the first two limitations (generalizability and UKB bias) with the following passage in the discussion:

“The UK Biobank is known to have ‘healthy volunteer bias’, as respondents tend to be healthier, more educated, and are more likely to own assets [71,72]. Various types of selection bias can occur in non-representative samples, impacting either internal (type 1) or external (type 2) validity. One benefit of a natural experimental design is that it protects against threats to internal validity from selection bias [43], design-based internal validity threats still exist, such as if volunteer bias differentially impacts individuals based on the cutoff for assignment. A more pressing limitation – in particular, for an education policy change – is our power to detect effects using a sample of higher-educated individuals. This is evident in our first stage analysis examining the percentage of 15-year-olds impacted by ROSLA, which we estimate to be 10% in neuro-UKB (Sup. Figure 2 & Sup. Table 2), yet has been reported to be 25% in the UK general population [41]. Our results should be interpreted for this subpopulation (UK, 1973, from 15 to 16 years of age, compliers) as we estimate a ‘local’ average treatment effect [73]. Natural experimental designs such as ours offer the potential for high internal validity at the expense of external validity.”

We further highlight this in the results section:

“Compliance with ROSLA was very high (near 100%; Sup. Figure 2). However, given the cultural and historical trends leading to an increase in school attendance before ROSLA, most adolescents were continuing with education past 15 years of age before the policy change (Sup Plot. 7b). Prior work has estimated 25 percent of children would have left school a year earlier if not for ROSLA 41. Using the UK Biobank, we estimate this proportion to be around 10%, as the sample is healthier and of higher SES than the general population (Sup. Figure 2; Sup. Table 2) 46–48.”

Healthy volunteer bias can create two types of selection bias; crucially participation *itself* can serve as a collider threatening internal validity (outlined in van Alten et al., 2024; https://academic.oup.com/ije/article/53/3/dyae054/7666749). Natural experimental designs are partially sheltered from this *major* limitation, as ‘volunteer bias’ would have to differentially impact individuals on one side of the cutoff and not the other – thereby breaking a primary design assumption of regression discontinuity. Substantial prior work (including this article) has not found any threats to the validity of the 1973 ROSLA (Clark & Royer 2010, 2013; Barcellos et al., 2018, 2023; Davies et al., 2018, 2023). While the Davies 2028 article did IP-weight with the UK Biobank sample, Barcellos and colleagues 2023 (and 2018) do not, highlighting the following “Although the sample is not nationally representative, our estimates have internal validity because there is no differential selection on the two sides of the September 1, 1957 cutoff – see Appendix A.”.

The second (more acknowledged & arguably less problematic) type of selection bias results in threats to external validity (aka generalizability). As highlighted in your first point; this is a large limitation with *every* natural experimental design, yet in our case, this is further amplified by the UK Biobank’s healthy volunteer bias. We have now attempted to highlight this limitation in the discussion passage above.

Point 3 – the inability to fully confirm design validity – is again, another inherent limitation of a natural experimental approach. That being said, extensive prior work has tested different predetermined covariates in the 1973 ROSLA (cited within), and to our knowledge, no issues have been found. The 1973 ROSLA seems to be one of the better natural experiments around (there was also a concerted effort to have an ‘effective’ additional year; see Clark & Royer 2010). For these reasons, we stuck with only testing the variables we wanted to use to increase precision (also offering new neuroimaging covariates that didn’t exist in the literature base). One additional benefit of ROSLA was that the cutoff was decided years later on a variable that happened (date of birth) in the past – making it particularly hard for adolescents to alter their assignments.

**Reviewer #3 (Recommendations for the authors):**
(1) FMRIB's preprocessing pipeline is mentioned. Does this include deconfounding of brain measures? Particularly, were measures deconfounded for age before the main analysis?

This is such a crucial point that we triple-checked, brain imaging phenotypes were not corrected for age (https://biobank.ctsu.ox.ac.uk/crystal/crystal/docs/brain_mri.pdf) – large effects of age can be seen in the global metrics; older individuals have less surface area, thinner cortices, less brain volume (corrected for head size), more CSF volume (corrected for head size), more white matter hyperintensities, and worse FA values. Figure 1 shows these large age effects, which are controlled for in our continuity-based RD analysis.

One’s date of birth (DOB) of course does not match perfectly to their age, this is why we included the covariate ‘visit date’; this interplay can now be seen in our updated SI Figure 1 (recommended in #3) which shows the distributions of visit date, DOB, and age of scan.

In a valid RD design covariates *should* not be necessary (as they should be balanced on either side of the cutoff), yet the inclusion of covariates does increase precision to detect effects. We tested this assumption, finding the effect of ‘visit date’ and its quadratic term to be not related to ROSLA (Sup. Table 1). This adds further evidence (specific to the UK Biobank sample) to the existing body of work showing the 1973 ROSLA policy change to not violate any design assumptions. Threats to internal validity would more than likely increase endogeneity and result in ‘false causal positive causal effects’ (which is not what we find).

(2) Despite the large overall sample size, I am wondering whether the effective number of samples is sufficient to detect a potentially subtle effect that is further attenuated by the long time interval before scanning. As stated, for the optimised bandwidth window (DoB 20 to 35 months around cut-off), N is about 5000. Does this mean that effectively about 250 (10%) out of about 2500 participants born after the cut-off were leaving school at 16 rather than 15 because of ROSLA? For the local randomisation analysis, this becomes about N=10 (10% out of 100). Could a power analysis show that these cohort sizes are large enough to detect a reasonably large effect?

This is a very valid point, one which we were grappling with while the paper was out for review. We now draw attention to this in the results and highlight this as a limitation in the discussion. While UKB’s non-representativeness limits our power (10% affected rather than 25% in the general population), it is still a very large sample. Our sample size is more in line with standard neuroimaging studies than with large cohort studies.

The novelty of our study is its causal design, while we could very precisely measure an effect of some phenotype (variable X) in 40,000 individuals. This effect is probably not what we think we are measuring. Without IP-weighting it could even have a different sign. But more importantly, it is not variable X – it is the thousands of things (unmeasured confounders) that lead an individual to have more or less of variable X. The larger the sample the easier it is for small unmeasured confounders to reach significance (Big data paradox) – this in no way invalidates large samples, it is just our thinking and how we handle large samples will hopefully change to a more casual lens.

(3) Supplementary Figure 1: A similar raincloud plot of date of birth would be instructive to visualise the distribution of subjects born before and after the 1957 cut-off.

Great idea! We have done this in Sup Fig. 1 for both visit date and DOB.

(4) p.9: Not sure about "extreme evidence", very strong would probably be sufficient.

As preregistered, we interpreted Bayes Factors using Jeffrey’s criteria. ‘Extreme evidence’ is only used once and it is about finding an associational effect of educational attainment on CSF (BF10 > 100). Upon Reviewer 1’s recommendation 7, we conducted eight replication samples (Sup. Figure 7 & 8) and have now added the following passage to the results:

“A post hoc replication of this associational analysis in eight additional 10-month cohorts spaced two years apart (Sup. Figure 7) indicates our preregistered report on the associational effect of educational attainment on CSF to be most likely a false-positive (Sup. Figure 8). Yet, the positive association between surface area and educational attainment is robust across the additional eight replication cohorts.”

(5) The code would benefit from a bit of clean-up and additional documentation. In its current state, it is not easy to use, e.g. in a replication study.

We have now further added documentation to our code; including a readme describing what each script does. The analysis pipeline used is not ideal for replications as the package used for continuity-based RD (RDHonest) initially could not handle covariates – therefore we manually corrected our variables after a discussion with Prof Kolesár (https://github.com/kolesarm/RDHonest/issues/7).

Prof Kolesár added this functionality recently and future work should use the latest version of the package as it can correct for covariates. We have a new preprint examining the effect of 1972 ROLSA on telomere length in the UK Biobank using the latest package version of RDHonest (https://www.biorxiv.org/content/10.1101/2025.01.17.633604v1). To ensure maximum availability of such innovations, we will ensure the most up-to-date version of this script becomes available on this GitHub link (https://github.com/njudd/EduTelomere).